# Physiological and Transcriptomic Analyses Reveal Exogenous Trehalose Is Involved in the Responses of Wheat Roots to High Temperature Stress

**DOI:** 10.3390/plants10122644

**Published:** 2021-12-01

**Authors:** Yin Luo, Yanyang Xie, Weiqiang Li, Maohuan Wei, Tian Dai, Zhen Li, Bozhi Wang

**Affiliations:** 1Instrument Sharing Platform of School of Life Sciences, East China Normal University, Shanghai 200241, China; xieyanyang0318@163.com (Y.X.); yuco_len964@163.com (M.W.); 18912845526@163.com (T.D.); 51201300111@stu.ecnu.edu.cn (Z.L.); sasoriforever@163.com (B.W.); 2Key Laboratory of Mollisols Agroecology, Northeast Institute of Geography and Agroecology, Chinese Academy of Sciences, Changchun 130102, China; liweqiang@iga.ac.cn

**Keywords:** wheat, trehalose, high temperature stress, transcriptomic analyses

## Abstract

High temperature stress seriously limits the yield and quality of wheat. Trehalose, a non-reducing disaccharide, has been shown involved in regulating plant responses to a variety of environmental stresses. This study aimed to explore the molecular regulatory network of exogenous trehalose to improve wheat heat tolerance through RNA-sequencing technology and physiological determination. The physiological data and RNA-seq showed that trehalose reduced malondialdehyde content and relative conductivity in wheat roots, and affecting the phenylpropane biosynthesis, starch and sucrose metabolism, glutathione metabolism, and other pathways. Our results showed that exogenous trehalose alleviates the oxidative damage caused by high temperature, coordinating the effect of wheat on heat stress by re-encoding the overall gene expression, but two wheat varieties showed different responses to high temperature stress after trehalose pretreatment. This study preliminarily revealed the effect of trehalose on gene expression regulation of wheat roots under high temperature stress, which provided a reference for the study of trehalose.

## 1. Introduction

High temperatures will occur more frequently in the future because of the greenhouse effect [1]. Additionally, it is predicted that the global average temperature will rise at least 4 °C by the end of this century [2]. High temperature affects all aspects of plants, such as inhibiting seed germination, limiting plant growth and reproduction, and reducing crop yield and quality [3]. When discussing the effects of high temperature on plants, people pay more attention to the aboveground tissue, and use compounds as initiators to improve the tolerance of different plants to abiotic stresses. For example, exogenous dopamine treatment improved the resistance of apple seedlings to drought [4]. Exogenous glutathione enhanced the tolerance of cucumber seedlings to heat stress by regulating photosynthesis, antioxidants, and penetrant systems to improve physiological adaptation [5]. However, the root is more sensitive to high temperature than the stem. High temperature not only causes the excessive accumulation of reactive oxygen species (ROS) and malondialdehyde (MDA) in the root, damaging cell components, but also leads to the loss of cell function [6]. Roots directly exposed to high temperature soil may regulate the response of stems to heat stress, inhibiting stem growth, photosynthesis, carbohydrate metabolism, and leading to leaf senescence [7]. Therefore, more attention should be paid to the study of roots.

Trehalose (α-d-glucopyranosyl-α-d-glucopyranoside) is a non-reducing disaccharide widely present in organisms [8,9]. In plants, trehalose-6-phosphate synthase (TPS) first synthesizes glucose-6-phosphate and uridine diphosphate glucose (UDP-glucose) into trehalose-6-phosphate (T6P), and then the phosphate group of T6P is removed under the action of trehalose-6-phosphate phosphatase (TPP) to produce trehalose, which is finally decomposed into glucose by trehalase.

Numerous studies have shown that trehalose can play a role in plant response to stresses. On the one hand, in the face of adversities, plants will actively increase the level of endogenous trehalose to deal with the adverse environment. This phenomenon was first verified in reviving plants. During dehydration, trehalose concentration in resurrected plants increased and acted as a protective agent for protein and cell membrane [10]. In addition, trehalose accumulation was detected in Arabidopsis, straw mushroom, wheat, cotton, Lucerne, and other plants under abiotic stress, which enhanced plant stress resistance [11,12,13,14]. 

On the other hand, exogenous trehalose can also reduce the negative environmental pressure faced by plants. Zhao et al. found that spraying trehalose can reduce the harmful effects of high temperature stress on Paeonia lactiflora pall by enhancing the antioxidant system, activating photosynthesis, and protecting cell structure [15]. Under cold stress, exogenous trehalose significantly increased the maximum quantum yield of PSII, the antioxidant enzyme activity, and endogenous H_2_O_2_ and NO levels, thus stimulating the response of melon to cold stress [16]. Tomato seedlings treated with 2 mmol/L trehalose can regulate the expression of genes related to ABA synthesis and enhance the salt tolerance of tomatoes [17]. Although the current research has shown that exogenous trehalose has excellent research value in improving plant stress resistance, there are few studies on the molecular mechanism of exogenous trehalose in response to abiotic stress. Further efforts are needed to better clarify the internal mechanism of trehalose in improving crop stress resistance.

Wheat (*Triticum aestivum* L.) is a typical cold season crop, and high temperature stress will shorten its growth period and significantly reduce the yield [18,19,20]. It is reported that the global wheat yield may decrease by 4.1–6.4% for every 1 °C increase in global temperature [21,22]. Therefore, to reduce economic losses and meet the needs of the global population, it is urgent to take adequate measures to improve the high temperature tolerance of wheat. Here we studied effects of root application of trehalose under high temperature stress on the physiological characteristics of Yangmai 18 (spring wheat) and Yannong 19 (winter wheat). Using high-throughput sequencing technology, we compared similarities and differences in the transcriptomic regulatory networks of the two varieties wheat root, trying to clarify the potential molecular adaptative mechanism of exogenous trehalose to improve the heat tolerance of wheat. 

## 2. Results

### 2.1. Trehalose and High Temperature Stress Affected Root Morphology of Wheat

Under high temperature stress, the roots of Yangmai 18 obviously turned yellow, the lateral roots decreased, and the root length became shorter (Figure 1A), while no significant change in root length in Yannong 19 was observed except that roots turned yellow (Figure 1B), indicating a better tolerance of Yannong 19 to high temperature. When the root length with trehalose pretreatment plus high temperature treatment was compared with those with only high temperature treatment, those in Yannong 19 reduced, while changes in Yangmai 18 were not observed (Figure 1). 

### 2.2. Trehalose Pretreatment Decreased Electrolyte Leakage and MDA Content of Wheat Roots under High Temperature Stress

In order to clarify whether exogenous trehalose improves the high temperature tolerance of wheat roots, we measured changes caused by trehalose in electrolyte leakage and MDA content. Compared with the control group under normal growth conditions, both the relative conductance (Figure 2A1,A2) and MDA content (Figure 2C1,C2) in Yangmai 18 and Yannong 19 increased significantly under high temperature stress, indicating that the cell membrane of both wheat roots was damaged under high temperature stress. However, exogenous trehalose-pretreated wheat roots showed lower electrolyte leakage (Figure 2A1,A2) and MDA content (Figure 2C1,C2) than those without trehalose pretreatment under high temperature stress. Compared with high temperature stress, exogenous trehalose significantly reduced the injury degree of Yangmai 18 (Figure 2B1), and the injury degree of Yannong 19 also showed a downward trend (Figure 2B2). These results demonstrated that trehalose pretreatment protected the cell membrane of wheat roots from high temperature-induced oxidative damage.

### 2.3. Transcriptome Sequencing and Quality Assessment

To explore the molecular mechanism of exogenous trehalose improving heat tolerance of wheat root, the cDNA library of each treatment was sequenced by Illumina Hiseq platform based on sequencing by synthesis (SBS) technology. The results showed that more than 42 million reads for each sample were produced (Table 1). After low-quality cDNA readings and splices were removed, a clean reading ratio of more than 91.5% per sample was obtained, indicating that it is enough for sequencing depth to cover the transcriptome of wheat roots (Table 1). In addition, the average base mass of Q30 in each treatment was greater than 93%, Q20 greater than 97%, and the GC content was between 48.35% and 54.34% (Table 1), indicating that the quality of transcriptome sequencing met the requirements of data analysis. When the Chinese Spring (Triticum dicoccoides-nudigl) genes were used as a reference for comparison, the comparison rates of each library to the reference genome were from 43.68% to 90.60% (Appendix A). This indicated that the sequencing data was reliable enough for subsequent bioinformatics analysis. Meanwhile, the R2 between biological repeated samples is more significant than 0.85, meaning that the repeatability between samples is good (Figure 3).

### 2.4. Analysis of Differentially Expressed Genes (DEGs)

According to |Log2 fold change| ≥ 1 and false discovery rate (FDR) < 0.05, the number of DEGs in each two groups was counted (Figure 4). The results showed that the maximum number of DEGs of the two varieties appeared in I_CK_vs_I_H and I_Tre_vs_I_TreH, respectively (Figure 4), indicating that Yangmai 18 is more sensitive to high temperature stress, and there are more genes involved in high temperature response. Under high temperature stress, the DEGs of Yangmai 18 were much more than those of Yannong 19 after trehalose pretreatment (I_H_vs_I_TreH and II_H_vs_II_TreH), indicating that the effect of exogenous trehalose on wheat under high temperature may be different in different varieties. Compared with Yannong 19, Yangmai 18 has more genes involved in trehalose response. In addition, I_CK_vs_I_Tre and II_CK_vs_II_Tre had the least number of differential genes, indicating that trehalose had little effect on wheat gene expression at normal temperature.

To verify the accuracy of transcriptome sequencing results, 12 genes were randomly selected for quantitative real-time PCR (qRT-PCR) verification (Figure 5). The results showed that the relative expression profiles of these genes were consistent with those of RNA sequencing.

### 2.5. Gene Ontology (GO) Analysis under High Temperature Stress

To investigate the functions of DEGs, we performed GO analyses. In total, 50 GO terms with the lowest q-value in the GO enrichment results were selected for analysis. In Yangmai 18, when trehalose was not added under high temperature stress (I_CK_vs_I_H), the functions of DEGs were mainly concentrated in sugar transport, DNA conformation change, protein folding, protein–DNA complex assembly, tubulin binding, and circadian rhythm (Figure 6A). After trehalose was applied, in the normal temperature group and high temperature group (I_Tre_vs_I_TreH), the functions of DEGs were mainly concentrated in protein–DNA complex subunit tissue, transition metal ion homeostasis, cytoplasmic ribosome, ribosomal subunit, and tubulin binding (Figure 6D).

In Yannong 19, under high temperature stress (II_CK_vs_II_H), the functions of DEGs were mainly concentrated in DNA conformational changes, monosaccharide metabolism, circadian rhythm, protein folding, tubulin binding, and histone binding (Figure 6E), and II_Tre_vs_II_TreH group showed that the functions of DEGs after trehalose application were mainly concentrated in DNA conformation change, circadian rhythm, cytoplasmic body, and tubulin binding (Figure 6H).

Under the normal growth conditions, effects of trehalose on DEGs in Yangmai 18 (I_CK_vs_I_Tre) were mainly manifested in cell ion homeostasis, positive regulation of catalytic activity, positive regulation of molecular function, and stress activated protein kinase signal cascade (Figure 6B), while in Yannong 19, the functions of DEGs in II_CK_vs_II_Tre were mainly concentrated in amino acid transport, detoxification, toxin metabolism, tricarboxylic acid biosynthesis, and glutathione transferase activity (Figure 6F).

Under high temperature stress conditions, effects of trehalose on DEGs in Yangmai 18 (I_H_vs_I_TreH) were mainly manifested in the process of amine synthesis and metabolism, regulation of chromatin tissue, NADH dehydrogenase complex, oxidoreductase complex, respiratory chain, histone binding, and transcriptional co-regulatory activity (Figure 6C). In Yannong 19, the functions of DEGs in II_H_vs_II_TreH were mainly concentrated in amine synthesis and metabolism, regulation of secondary metabolism, amino acids, gibberellin reaction, tricarboxylic acid synthesis and metabolism, and oxidoreductase activity (Figure 6G).

### 2.6. Kyoto Encyclopedia of Genes and Genomes (KEGG) Pathways Enrichment Analysis under High Temperature Stress

To further explore the metabolic regulatory network of wheat under high temperature stress, the identified DEG was analyzed by KEGG metabolic pathway. The top 20 most obviously enriched pathways are shown in Figure 7. In Yangmai 18, when trehalose was not root-applied, the DEGs at normal temperature and high temperature group (I_CK_vs_I_H) were mainly enriched in phosphatidylinositol signaling system, phenylpropanoid biosynthesis, nitrogen metabolism, glycolysis/gluconeogenesis, DNA replication, biosynthesis of amino acids (Figure 7A). When trehalose was applied, the DEGs in I_Tre_vs_I_TreH were basically consistent with those in I_CK_vs_I_H, except for 7 different KEGG pathways, and the DEGs in I_Tre_vs_I_TreH were mainly enriched in ribosome, protein processing in endoplasmic reticulum, phenylpropane biosynthesis, amino acid biosynthesis (Figure 7D). Under the normal growth conditions, the enrichment degree of DEGs in the trehalose-free group and the trehalose pretreatment group (I_CK_vs_I_Tre) was light and mainly enriched in protein processing in endoplasmic reticulum, MAPK signaling pathway, and endocytosis (Figure 7B). Under the same high temperature, the DEGs in the trehalose-free group the trehalose pretreatment group (I_H_vs_I_TreH) were mainly enriched in sulfur metabolism, RNA degradation, phosphatidylinositol signaling system, glycolysis/gluconeogenesis, carbon metabolism, biosynthesis of amino acids, autophagy-other (Figure 7C).

In Yannong 19, in the absence of trehalose, the DEGs in normal temperature and high temperature group (II_CK_vs_II_H) were mainly enriched in phenylpropanoid biosynthesis, glycolysis/gluconeogenesis, glutathione metabolism, galactose metabolism, biosynthesis of amino acids, and ABC transporters (Figure 7E). Compared with II_CK_vs_II_H, the DEGs in II_Tre_vs_II_TreH were mainly enriched in nitrogen metabolism, carbon metabolism, secondary metabolite biosynthesis (Figure 7H). At the normal temperature, the DEGs in II_CK_vs_II_Tre were mainly enriched in vitamin B6 metabolism, phenylpropane biosynthesis, metabolic pathway, glutathione metabolism, and biosynthesis of secondary metabolites (Figure 7F). At high temperature, the DEGs in the trehalose-free group and the trehalose pretreatment group (II_H_vs_II_TreH) were mainly enriched in plant hormone signal transduction, phenylpropanoid biosynthesis, metabolic pathways, MAPK signaling pathway-plant, glutathione metabolism, glyoxylic acid, and dicarboxylic acid metabolism, biosynthesis of secondary metabolites (Figure 7G). Similar to the results of Yangmai 18, the role of trehalose in Yannong 19 is also significantly different due to different environments.

### 2.7. Trehalose Responsive Transcription Factors (TFs) Analysis

TFs also play a key role in trehalose response and abiotic stress via gene regulatory networks. According to fold change value, the DEGs in I_H_vs_I_TreH and II_H_vs_II_TreH were analyzed to screen out the trehalose responsive TFs in the two strains. A total of 13 differentially expressed TFs genes were found, including 10 families such as *NAC* (TraesCS2D02G378800), *MYB* (TraesCS3A02G108000), *WRKY* (TraesCS6A02G146900), and so on. (Appendix A). All of these TFs were previously characterized as regulating heat resistance. Among them, the expression of 5 genes including *MYB* (TraesCS3A02G108000) and *AP2/ERF-ERF* (TraesCS5B02G481300) was downregulated in two varieties, the expression of 8 genes including *NAC* (TraesCS2D02G378800), *bHLH* (TraesCS4B02G331900) and *WRKY* (TraesCS6A02G146900) was upregulated in Yangmai 18 and downregulated in Yannong 19 (Figure 8).

### 2.8. Trehalose Affected Expression Profiling of Genes Involved in Metabolic Pathway and Defense Response under High Temperature Stress

In Yangmai 18, a total of 362 DEGs involved in phenylpropanoid biosynthesis (209 DEGs) and starch and sucrose metabolism (153 DEGs) were identified. In Yannong 19, a total of 37 DEGs involved in phenylpropanoid biosynthesis (28 DEGs) and starch and sucrose metabolism (9 DEGs) were identified. To further study the effects of trehalose pretreatment on changes of metabolic pathways when plants suffered from heat stress, we selected 17 DEGs from phenylpropane biosynthesis and starch and sucrose metabolism from Yangmai 18 and Yannong 19 for expression profile analysis (Figure 9 and Appendix A).

Our data showed that the expression trend of most DEGs involved in phenylpropane biosynthesis in the two wheat varieties was the same (Figure 9). Phenylalanine ammonia-lyase (PAL), trans-cinnamate 4-hydroxylase (C4H), and 4-coumarate-CoA ligase (4CL) are the key enzymes that catalyze the first three steps of phenylpropane biosynthesis and provide a material basis for the downstream specific synthesis pathway. Under high temperature stress, trehalose pretreatment induced upregulation of genes encoding PAL (TraesCS1A02G037800, TraesCS2A02G381100, TraesCS2B02G224300, TraesCS2D02G377500) and 4CL (TraesCS2A02G145800) (Figure 9 and Appendix A). The expression pattern of enzymes related to lignin biosynthesis is shown in Figure 9. In Yangmai 18, the genes encoding cinnamoyl-CoA reductase (CCR) and cinnamyol alcohol dehydrogenase (CAD) are downregulated, while the genes encoding peroxidase (*POD*) are mostly upregulated. The expression patterns of *CAD* (TraesCS7B02G282100, TraesCS1A02G424100) and *POD* (TraesCS2B02G124800, TraesCS4D02G116600, TraesCS7B02G328400, TraesCS7D02G420500) in Yannong 19 were opposite to those in Yangmai 18. Beta-glucosidase (BGL; TraesCS5D02G399100) is considered to play a key role in lignification, but the gene encoding it was downregulated in both varieties. In addition, the gene encoding scopoletin glucosyltransferase (TOG1; TraesCS5A02G433200) was downregulated in Yangmai 18 and upregulated in Yannong 19.

In the starch and sucrose metabolic pathways, endoglucanase (EG) is one of the genes encoding glycolytic enzymes, our study found that the gene encoding EG (TraesCS6A02G312400) was upregulated in Yangmai 18 and downregulated in Yannong 19 (Figure 9 and Appendix A). BGL (TraesCS5D02G399100) and glucan endo-1,3-beta-glucosidase 1/2/3 (GN1_2_3; TraesCS5A02G338400) were downregulated in both wheat varieties (Figure 9 and Appendix A).

This study found that trehalose also affected glutathione metabolism. A total of 109 DEGs involved in glutathione metabolism in Yangmai 18 and 12 DEGs in Yannong 19 were identified. We selected 2 DEGs encoding glutathione S-transferase (GST; TraesCS2A02G045400, TraesCS5D02G027500) in H and TreH groups for expression profile analysis (Figure 9 and Appendix A). In Yannong 19, after trehalose pretreatment under high temperature stress, the expression level of GST was significantly higher than that without trehalose pretreatment, and the expression levels of 5-oxoprolinase (OPLAH; TraesCS6D02G288100) and 6-phosphogluconate dehydrogenase (6-PGD; TraesCS1A02G357200) decreased. In Yangmai 18, there were more DEGs involved in glutathione metabolism. The expression trend of GST and 6-PGD (TraesCS7D02G030000) was the same as that in Yannong 19, while the gene encoding *OPLAH* (TraesCS1B02G197800) was upregulated, and the gene encoding gamma-glutamylcyclotransferase (GGCT; TraesCS2A02G316700, TraesCS6A02G068300, TraesCS6D02G065900) was decreased (Appendix A). The results showed that trehalose could participate in regulating ROS dynamic balance under heat stress.

Plants defend stress by expressing specific genes and the synthesis of stress proteins. Under high temperature stress, compared with non-trehalose treatment, in trehalose pretreated wheat roots, the expression level of some genes encoding pathway-related proteins decreased and heat shock proteins (HSPs) increased (Appendix A) indicating that trehalose alleviated heat stress.

Heat stress leads to protein misfolding and denaturation, and autophagy can remove proteins and protein aggregates with misfolding and denaturation. Autophagy is highly regulated by a series of different autophagy related (ATG) proteins. Under heat stress, in Yangmai 18, a total of 39 DEGs involved in autophagy pathway were identified, while in Yannong 19, only 1 DEG involved in autophagy pathway was identified (Appendix A). External application of trehalose under high temperature stress, the genes encoding ATG1, ATG2, ATG9, ATG13, ATG6, ATG7, and ATG18 were upregulated in Yangmai 18, and the genes encoding target of rapamycin complex subunit LST8 (MLST8) were downregulated (Appendix A). In Yannong 19, only 1 DEG encoding MLST8 was upregulated (Appendix A).

## 3. Discussion

Consistent with previous studies that trehalose could effectively alleviate the inhibitory effect of abiotic stress on plants [15,16,17], the present study showed that the application of trehalose can maintain the phenotypic properties of wheat roots, endow them with more robust vitality, and help them resist heat stress, but this characteristic is more obvious in Yangmai 18 than Yannong 19 (Figure 1). During abiotic stress (ultraviolet and heat stress, etc.), ROS levels will rise sharply, affect proteins and lipids, and lead to cell damage and death [23]. ROS scavenging mechanism plays an essential role in protecting plants from stress damage [24]. Trehalose pretreatment also reduced MDA content and relative conductivity in roots and helped alleviate oxidative damage caused by high temperature stress (Figure 2), which may be caused by the increase in glutathione metabolism in the ROS scavenging pathway (Figure 9).

RNA-seq data showed that trehalose significantly affected gene expression regulation. In the H vs. TreH group, 15403 DEGs (8042 upregulated and 7361 downregulated) in Yangmai 18 and 749 DEGs (234 upregulated and 515 downregulated) in Yannong 19 were found, indicating that there are more genes involved in trehalose response in Yangmai 18 (Figure 4). The response of numerous genes to trehalose pretreatment showed that the response of trehalose to high temperature stress was extraordinarily complex. We speculated that the response DEGs were different due to the difference in stress resistance between the two varieties of wheat. Combined with physiological data, we considered that Yangmai 18 was more sensitive to environmental changes, and Yannong 19 had stronger resistance to high temperature. GO analysis showed that most DEGs were related to cell metabolism and biological regulation (Figure 6C,G). KEGG enrichment analysis showed that amino acid metabolism (such as arginine, proline, and phenylalanine) is dominant in the primary metabolism of wheat affected by exogenous trehalose (Figure 7C,G). These amino acids may be used as compatible penetrants, precursors of secondary metabolites or storage forms of organic nitrogen, and play a protective role in plants in response to abiotic stress [25].

RNA-seq analysis data also showed that exogenous trehalose could activate autophagy in response to high temperature stress, however, the results of the two varieties were quite different (Figure 9). Plant autophagy can help plant cells recover damaged or unwanted proteins and organelles, to maintain the continuous supply of nutritional energy. This process is highly regulated by a series of different ATG proteins. Our KEGG results showed that trehalose induced the upregulation of genes encoding ATG protein in Yangmai 18. At the same time, there was no change in genes encoding ATG protein in Yanlong 19 (Appendix A). This result is consistent with previous studies, in various plants and algae, including Arabidopsis, Chlamydomonas, rice, and wheat, the expression level of many ATG genes increases under abiotic stress [26]. Moreover, recent studies on tomato have shown that WRKY33 transcription factor may be a regulator of autophagy during heat stress [27]. In this study, the expression of WRKY33 in Yangmai 18 roots increased under the action of trehalose but decreased in Yannong 19 (Appendix A). Therefore, we can speculate that trehalose can regulate the expression of ATG through the role of WRKY33 transcription factor, leading to autophagy and improving the ability of plants to resist high temperature.

When plants are subjected to abiotic stress, many metabolic processes will also change to respond to stress [28]. The comprehensive analysis of DEGs of phenylpropane biosynthesis showed that trehalose pretreatment increased phenylpropane biosynthesis and provided a material basis for subsequent different phenylpropane metabolites, including flavonoids, lignin, and anthocyanins (Figure 9 and Appendix A). In addition, it was found from trehalose-responsive candidate genes that carotenoid 9,10(9′,10′)-cleavage dioxygenase 1 (CCD1) was upregulated in both varieties, helping carotenoid oxidize to produce substances such as abscisic acid [29]. Our RNA-seq results also showed that trehalose induced the biosynthesis of small molecules such as glucose and fructose under heat stress (Figure 9 and Appendix A). These small molecules may be used as intermediate substances to supply energy metabolism under heat stress to maintain metabolic dynamic balance and may also be related to osmotic regulation [30]. Our results are consistent with research by Rezaul et al. [31] that exogenous abscisic acid maintains carbon balance and energy balance by affecting sucrose metabolism, thus improving the heat tolerance of rice. These results suggest that the application of trehalose in wheat plants has changed the transcriptional recoding of metabolic pathways, and there may be a complex regulatory network initiated by trehalose, which can help plants quickly respond to heat stress or ensure plant growth and development under adverse conditions.

## 4. Conclusions

In conclusion, our results show that exogenous trehalose can help plants perform better under heat stress, alleviate the oxidative damage caused by high temperature, and start gene activation, which helps to eliminate excessive ROS, induce the accumulation of defense proteins, and cause changes in metabolic process. At the same time, we also found that wheat varieties with different heat tolerance had different responses to high temperature stress after trehalose pretreatment. In Yangmai 18 higher heat sensitivity and more DEGs were found than those in Yannong 19. In addition, in Yangmai 18, trehalose can regulate the expression of ATG through WRKY33 transcription factor, resulting in autophagy, which plays a role in improving wheat heat tolerance, but the same process was not found in Yannong 18 (Figure 10). These changes induced by trehalose trigger the development of plant heat tolerance and physiological dynamic balance there by protecting plants for better growth and development. However, whether this is the case in the leaves requires further research.

## 5. Materials and Methods

### 5.1. Plant Materials and Stress Treatment

Two different wheat varieties, namely Yangmai 18 (spring type) and Yannong 19 (winter type), were selected as experimental materials. First, the complete and full wheat seeds were selected, washed with water for 3–5 times, soaked in water in the beaker for 12 h, and allowed to germinate at room temperature. Then, two wheat cultivars were cultivated in the growth chamber (23 °C, 13 h light/11 h dark cycle, 120 μmol m^−2^ s^−1^ light intensity) in plastic pots hydroponically cultured with 1/2 Hogland solution. When the seedlings were in 2 true leaves and a terminal bud stage, they were pretreated with 0.5 mM trehalose (Tre), and the other group used 1/2 Hogland solution as control (CK). After 3 days of pretreatment, high temperature treatment group seedlings were transferred to the light incubator at heat stress condition (42 °C, 13 h light/11 h dark cycle, 120 μmol m^−2^ s^−1^ light intensity) for 24 h (H). All treatments are shown in Table 2. Water or nutrient solution was refreshed and ventilated once a day. The wheat roots from group CK, Tre, H, and TreH were collected for the following physiological experiment and transcriptome analysis. All the physiological experiments were repeated 3 times with 3 replicates each. RNA-Seq was performed with three biological replications in every sample.

### 5.2. Physiological Measurement of Wheat Roots under High Temperature Stress

MDA content was determined according to Zheng et al. [32]. First, 0.3 g wheat roots were crushed, added to 3 mL 10% (*w*/*v*) trichloroacetic acid (TCA) solution, and centrifuged at 5000× *g* for 10 min at 4 °C. Then, 1 mL supernatant mixed with 1 mL 0.67% (*w*/*v*) thiobarbituric acid (TBA) was heated for 15 min in a bath of boiling water. After cooling, the mixed solution was centrifuged at 5000× *g* for 10 min at 4 °C, and the absorbance of the supernatant was measured at 450 (OD450), 532 (OD532), and 600 nm (OD600), respectively. The MDA concentration and MDA content of the sample was calculated according to the following formula.

MDA concentration (μmol·L^−1^) = 6.45 × (OD532 − OD600) − 0.56 × OD450; MDA content (μmol·L^−1^) = (MDA concentration × extract volume)/Fresh weight of plant tissue.

Electrolyte leakage was determined according to Hafez et al. [33]. In total, 0.2 g root material was washed in deionized water and then surface moisture was wiped off. The initial conductivity (S1) was measured with a conductivity meter (Mettler Toledo, Greifensee, Switzerland) after subjecting the samples to incubation at 25 °C in 10 mL deionized water for 2 h and concussion frequently. Then, the samples were heated for 15 min in boiling water. After the samples cooled to room temperature, and the final conductivity (S2) was measured. The relative conductivity and injury degree of the sample was calculated according to the following formula.

Relative conductance = S1/S2; Injury degree (%) = (L_t_ − L_CK_)/(1 − L_CK_) × 100.

In the formula: L_t_—relative conductance of treated roots;

L_CK_—relative conductance of control roots.

### 5.3. RNA Sequencing and Annotation

RNA extraction, RNA detection, library construction, and sequencing were performed as previously described [34]. The samples of CK, Tre, H, and TreH from two wheats were selected for total RNA extract, using the Trizol reagent (Invitrogen, Waltham, MA, USA). The quantity and quality of RNA were determined by NanoDrop 2000 spectrophotometer (NanoDrop Technologies, Wilmington, DE, USA) and an Agilent 2100 Bioanalyzer (Agilent Technologies, CA, USA), respectively. The cDNA library was prepared for sequencing using the NEBNext Ultra RNA Library Prep Kit. RNA-seq analysis was conducted using an Illumina HiSeq platform at the MetWare company (Wuhan, China).

### 5.4. Transcriptomic Data Analysis

Screening of DEGs: the DEG sets between the two biological conditions were obtained by using DESeq2 [35,36]. The read counting of genes was realized by feature Counts [37]. After the difference analysis, we used the Benjamin–Hochberg method to correct the hypothesis test probability (*p*-value) but got the false discovery rate (FDR). The screening conditions for differential genes were |log2fold change| ≥1 and FDR < 0.05.

GO analysis: GO functions analysis by the OmicShare tools (http://www.omicshare.com/tools, accessed on 16 November 2021). GO term significant enrichment analysis was based on GO term in GO database (http://geneontology.org/, accessed on 17 November 2021). Hypergeometric test was used to find out the GO terms that were significantly enriched in differentially expressed genes compared with the whole genome background. The enrichment analysis results were presented in the form of a column chart.

KEGG analysis: KEGG pathway analysis by the OmicShare tools (http://www.omicshare.com/tools/, accessed on 16 November 2021). Pathway significant enrichment analysis takes the pathway in the KEGG database (https://www.kegg.jp/, accessed on 17 November 2021) as a unit, applies a hypergeometric test to find out the paths that are significantly enriched in the differentially expressed genes compared with the whole genome background, and obtains that the KEGG enrichment results are presented in the form of scatter diagram.

### 5.5. Validation of DEGs by qRT-PCR

According to the manufacturer’s protocol, total RNA was extracted from wheat roots using uniq-10 Trizol column total RNA Extraction Kit (Sangon Biotech, Shanghai, China), and cDNA was synthesized by two-step qRT-PCR premix (degenomic) kit (novozan biotechnology, Nanjing, China). qRT-PCR analysis was performed with SYBR Green chimeric fluorescent quantitative PCR kit (Novozan Biotechnology, Nanjing, China). Ubiquitin was used as the internal reference gene, and qRT-PCR was performed by Bio-Rad CFX-96 real-time PCR detection system (Bio-Rad, Hercules, CA, USA). A comparative Ct method (2-ΔΔCt) of relative quantification was used to evaluate the quantitative variation. The specific primers of 12 DEGs were designed using Primer 3 plus and then synthesized by Sangon Biotech (Shanghai, China) (Appendix A).

### 5.6. Statistical Analysis

Statistical analysis was performed using GraphPad Prism 8 version 8.0.2 software (Graph Pad Sofeware, San Diego, CA, USA). Two-way ANOVA and Tukey’s multiple comparison tests were used for statistical analysis. Data were presented as mean ± SE. A *p*-value of ≤0.05 was considered as statistically significant. * *p* ≤ 0.05, ** *p* ≤ 0.002, *** *p* ≤ 0.001.

## Figures and Tables

**Figure 1 plants-10-02644-f001:**
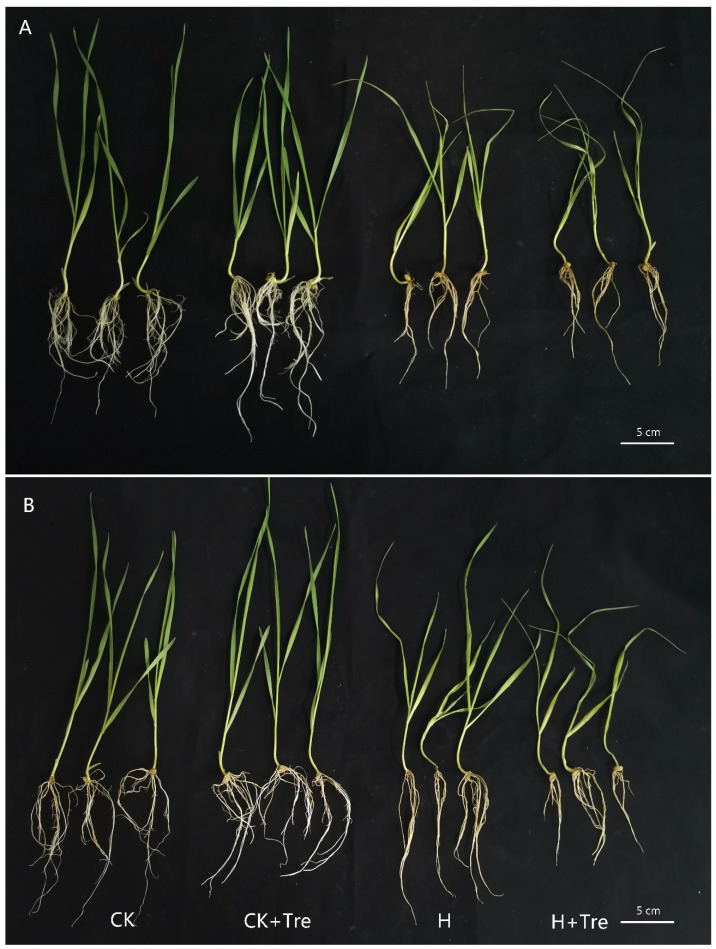
Phenotypic changes of wheat root induced by exogenous trehalose under high temperature stress. (**A**) Yangmai 18. (**B**) Yannong 19. CK: control group; H: high temperature stress treatment for 24 h; Tre: trehalose pretreatment for 3 d; TreH: trehalose pretreatment for 3 d was followed by high temperature stress treatment for 24 h.

**Figure 2 plants-10-02644-f002:**
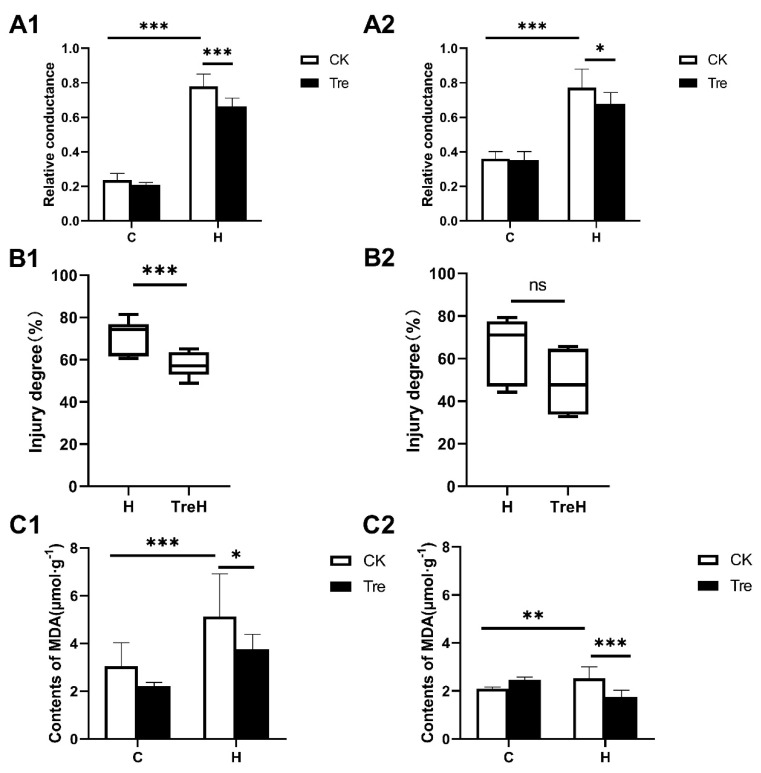
Trehalose reduces membrane lipid peroxidation of wheat roots under high temperature stress. (**A1**) The relative electrical conductivity of Yangmai 18. (**A2**) The relative electrical conductivity in Yannong 19. (**B1**) Injury degree of Yangmai 18. (**B2**) Injury degree of Yannong 19. (**C1**) The MDA content in Yangmai 18. (**C2**) The MDA content in Yannong 19. C: non-stressed seedlings, H: 42 °C stressed seedings for 24 h, CK: control group with Hoagland solution, Tre: trehalose pretreatment group, TreH: 42 °C stressed seedlings with 0.5 mM trehalose. Data were presented as mean ± standard error (SE). A *p*-value of ≤ 0.05 was considered as statistically significant. * *p* ≤ 0.05, ** *p* ≤ 0.002, *** *p* ≤ 0.001.

**Figure 3 plants-10-02644-f003:**
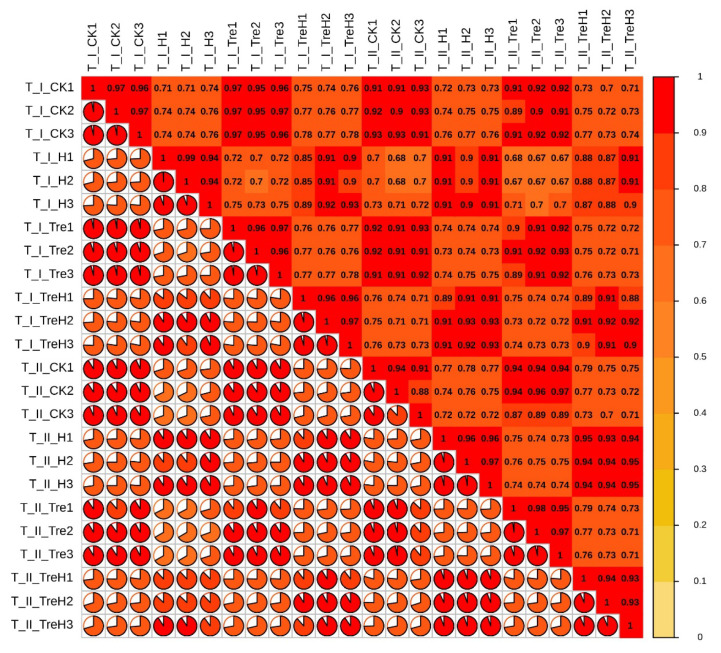
Correlation thermography. I: Yangmai 18; II: Yannong 19. CK: control; Tre: 0.5 mM trehalose pretreatment for 3 d; H: high temperature treatment for 24 h; TreH: 0.5 mM trehalose pretreatment for 3 d then high temperature treatment for 24 h.

**Figure 4 plants-10-02644-f004:**
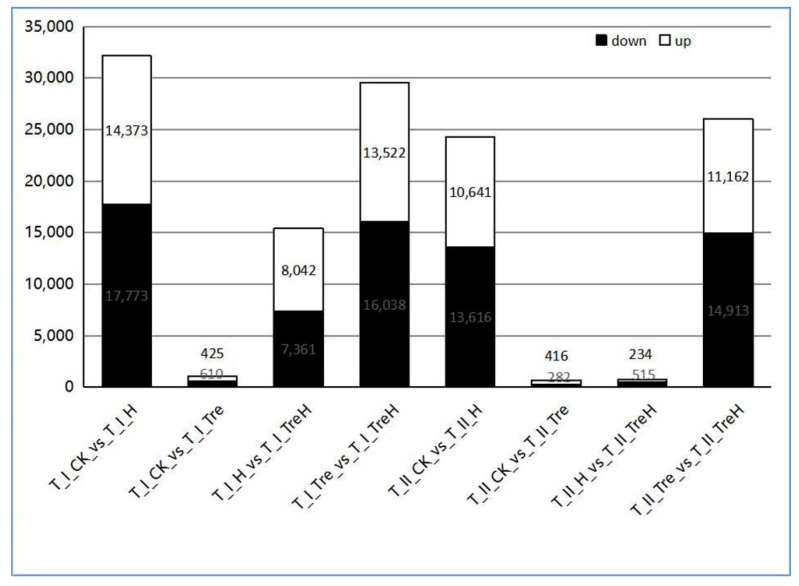
Differential expressed gene statistics among the eight comparison groups.

**Figure 5 plants-10-02644-f005:**
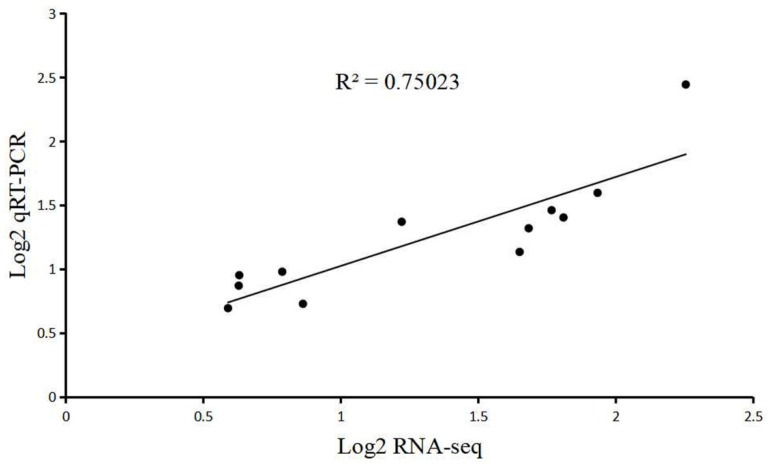
Correlation of expression levels of 12 genes between RNA-seq and qRT-PCR. The log2 qRT-PCR (*y*-axis) was plotted against log2 RNA-seq (*x*-axis).

**Figure 6 plants-10-02644-f006:**
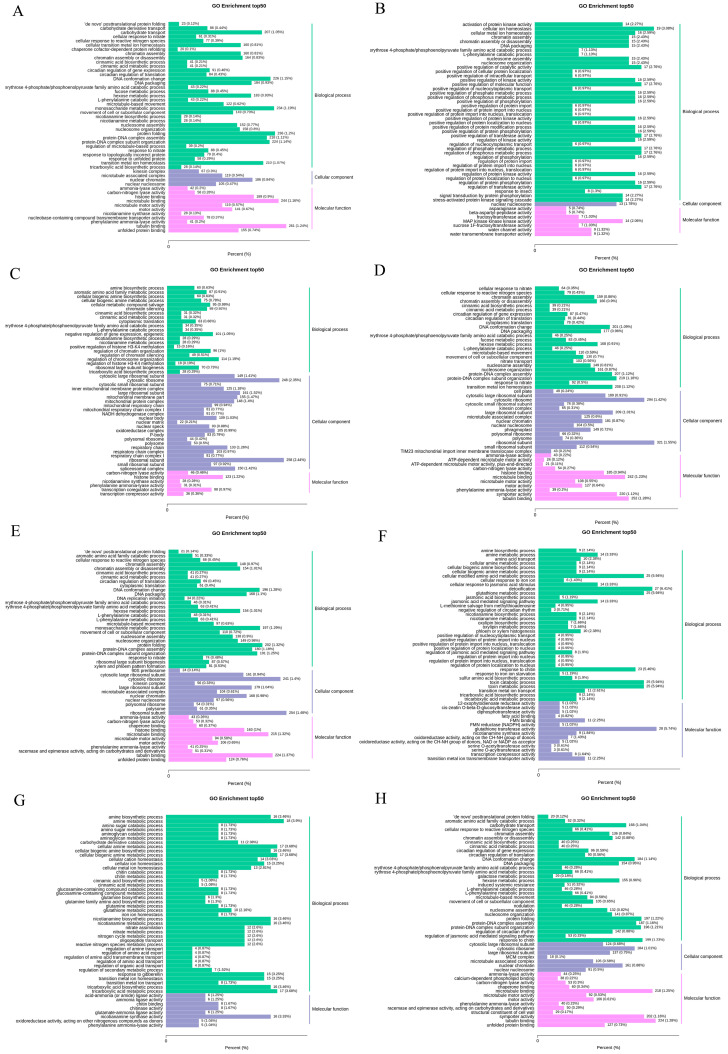
GO enrichment analysis of differential genes. The abscissa represents the ratio of the genes annotated to the entry to the total number of genes annotated, and the ordinate represents the name of the go entry. The label on the right side of the graph represents the category to which the go entry belongs. (**A**) I_CK_vs_I_H; (**B**) I_CK_vs_I_Tre; (**C**) I_H_vs_I_TreH; (**D**) I_Tre_vs_I_TreH; (**E**) II_CK_vs_II_H; (**F**) II_CK_vs_II_Tre; (**G**) II_H_vs_II_TreH; (**H**) II_Tre_vs_II_TreH.

**Figure 7 plants-10-02644-f007:**
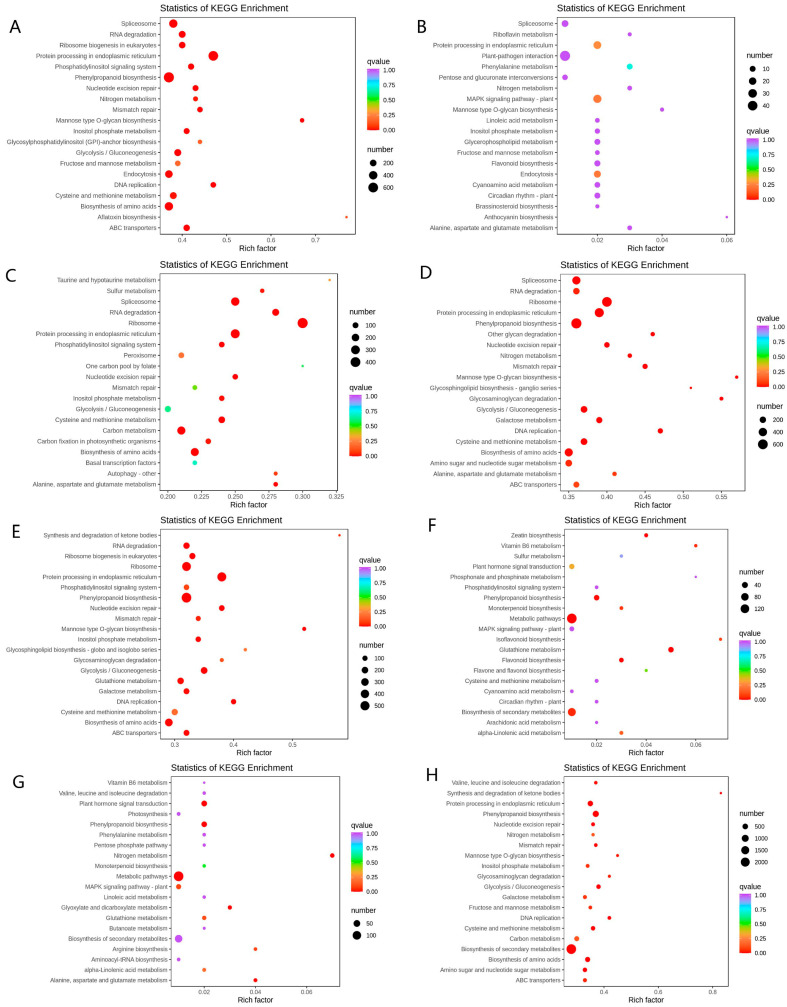
KEGG enrichment analysis of DEGs. KEGG enrichment is measured by rich factor, q-value, and the number of genes enriched in this pathway. (**A**) I_CK_vs_I_H; (**B**) I_CK_vs_I_Tre; (**C**) I_H_vs_I_TreH; (**D**) I_Tre_vs_I_TreH; (**E**) II_CK_vs_II_H; (**F**) II_CK_vs_II_Tre; (**G**) II_H_vs_II_TreH; (**H**) II_Tre_vs_II_TreH.

**Figure 8 plants-10-02644-f008:**
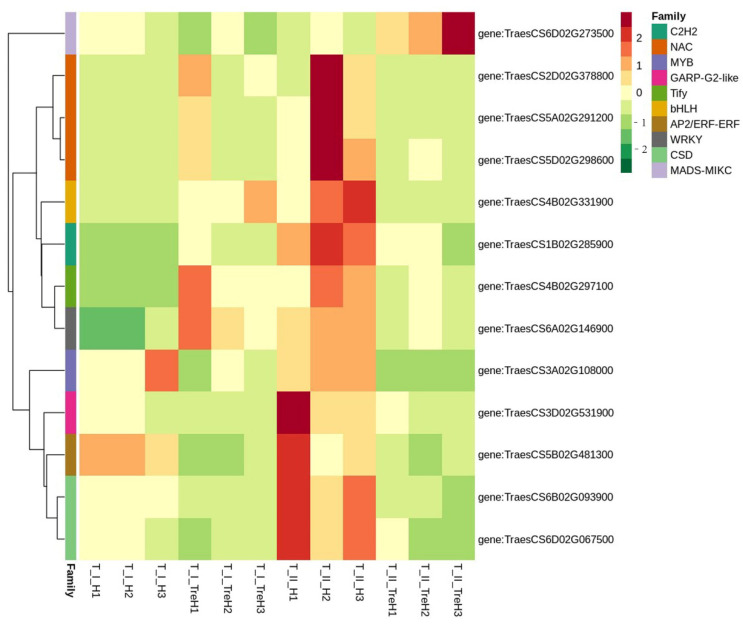
Heatmap was constructed based on FPKM expression values of TF.

**Figure 9 plants-10-02644-f009:**
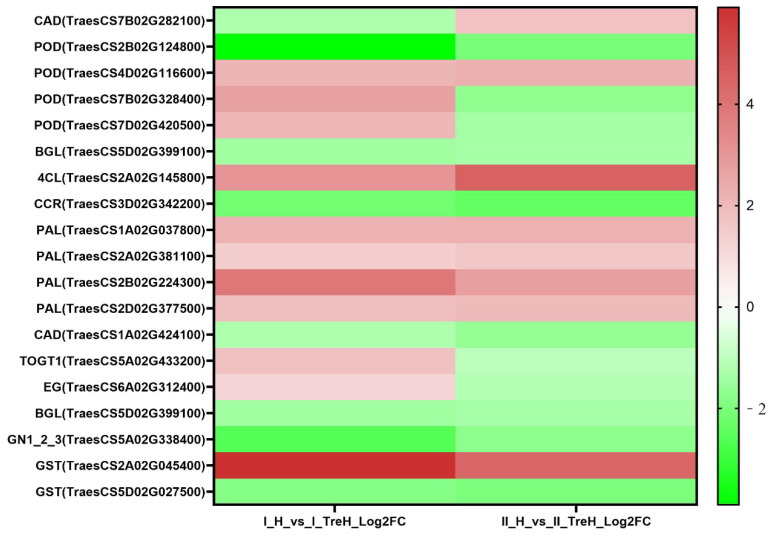
Heatmap representing expression dynamics of genes involved in metabolic pathway and defense response.

**Figure 10 plants-10-02644-f010:**
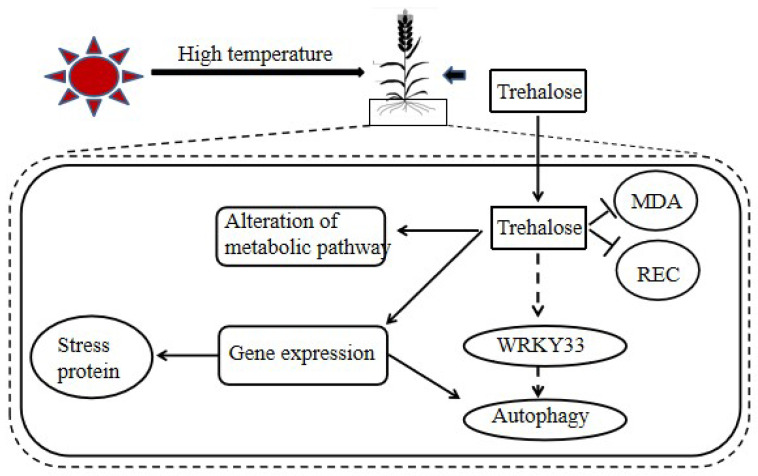
Model diagram of molecular mechanism of trehalose improving heat tolerance of wheat.

**Table 1 plants-10-02644-t001:** Transcriptome sequencing quality of different treatments.

Sample	Raw Reads	Clean Reads	Q20 (%)	Q30 (%)	GC Content (%)
I_CK1	43154458	40616236	98.4	95.09	53.33
I_CK2	42175284	39889014	98.52	95.49	53.12
I_CK3	51545892	49152868	98.47	95.34	53.04
I_H1	59683644	56870246	98.12	94.5	52.66
I_H2	60376404	57692984	97.65	93.56	52.88
I_H3	55557484	52700524	98.17	94.59	52.76
I_Tre1	45872598	42667542	98.53	95.46	51.24
I_Tre2	45823578	42410386	98.65	95.69	49.96
I_Tre3	48468438	45010626	98.7	95.83	50.44
I_TreH1	46768072	44011390	98.42	95.1	51.75
I_TreH2	48790306	44788578	98.63	95.78	50.79
I_TreH3	45721444	42171540	98.29	94.94	50.15
II_CK1	47916468	45472940	98.43	95.11	51.65
II_CK2	45095704	42159000	98.65	95.75	54.34
II_CK3	60113568	55562728	98.45	95.24	49.62
II_H1	63208358	60397222	97.98	94.21	52.5
II_H2	44846380	43018344	98.46	95.48	53.44
II_H3	45002306	41747874	98.3	95.2	53.41
II_Tre1	48842718	45208836	98.51	95.38	52.17
II_Tre2	46050176	43514846	98.63	95.67	52.63
II_Tre3	51242702	47433328	98.58	95.53	52.16
II_TreH1	60518452	56670044	98.2	94.76	51.57
II_TreH2	81493576	76716902	98.09	94.32	48.52
II_TreH3	96680092	89716602	97.97	93.98	48.35

I: Yangmai 18; II: Yannong 19. CK: control; Tre: 0.5 mM trehalose pretreatment for 3 d; H: high temperature treatment for 24 h; TreH: 0.5 mM trehalose pretreatment for 3 d then high temperature treatment for 24 h.

**Table 2 plants-10-02644-t002:** Treatment of wheat experimental groups.

Group	0.5 mM Trehalose Pretreatment	High Temperature Treatment
I_CK		
I_Tre	✓	
I_H		✓
I_TreH	✓	✓
II_CK		
II_Tre	✓	
II_H		✓
II_TreH	✓	✓

I: Yangmai 18. II: Yannong 19. CK: control; Tre: 0.5 mM trehalose pretreatment for 3 d; H: high temperature treatment for 24 h; TreH: 0.5 mM trehalose pretreatment for 3 d then high temperature treatment for 24 h.

## Data Availability

The datasets supporting the results of this article are available at the Sequence ReadArchive (SRA) database of National Center for Biotechnology Information (NCBI; https://www.ncbi.nlm.nih.gov/) under project accession number PRJNA762581.

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
