# Peer review of "Physiological and Transcriptomic Analyses Reveal Exogenous Trehalose Is Involved in the Responses of Wheat Roots to High Temperature Stress"

_plants, 2021, doi:10.3390/plants10122644_

Round 1

Reviewer 1 Report

The manuscript entitled with 'Physiological and transcriptomic analyses reveal exogenous trehalose is involved in the responses of wheat roots to high temperature stress' by Luo et al, presented phenotypic and transcriptomic responses of exogenous trehalose treatment under heat stress in wheat roots for two varieties. In general, the paper is well written and organized, however, minor revision is needed before publication.

Comments:

  1. From the results in Figure 1 and 2 as well as the identified DEG number, it seems exogenous trehalose didn't affect Yannong 19 much for heat resistance, or even has some negative effects, such as shortened root length and elevated MDA between CK and CK+Tre shown in Figure 1 and 2. Therefore, it's unreasonable to simply conclude that 'Our results showed that exogenous trehalose can help wheat roots perform better under heat stress' in abstract or 'the present study showed that the application of trehalose can maintain the phenotypic properties of wheat roots, endow them with more robust vitality and help them resist heat stress (Fig. 1)' in discussion (Line 314-316). Actually, the expression 'At the same time, we also found that wheat varieties with different heat tolerance had different responses to high temperature stress after trehalose pretreatment' (in discussion, Line 376-378) is more accurate and should be added to Abstract and conclusion sections.
  2. Figure 6 can not be clearly visualized. The author can either improve image quality or move it to supplementary material since sufficient description was included in corresponding text.
  3. In Figure 8 legend, isn’t 'FPKM' should be 'read counts' if used DESeq2 as mentioned in Material and methods?
  4. The explanation of 'I' and 'II' should be included in legend of Table 1 and Figure 3.
  5. Figure '9A' and '9B' were mentioned in manuscript but not showed in Figure 9. In addition, I suggest to use gene name instead of gene ID in Figure 9 since the gene names were talked throughout the manuscript. Gene ID could be added after gene name in parentheses in text.
  6. Raw RNA-seq data should be deposit to public database.

Author Response

Please see the attachment. Thank you very much for your review.

Reviewer 2 Report

I believe that there is data which contributes to the community. I have, however, major concerns regarding the MS.

  1. There are a number of figures that are not necessary. Figures can be combined and should be clearer and more concise. For Instance, Figure 9, 7 are necessary? Is it possible to combine???
  2. Please remove too old references and replace with the new ones. Please see these references to see if they help to improve our MS.

---Transcriptome pathways unique to dehydration tolerant relatives of modern wheat

Functional & integrative genomics 9 (3), 377-…

2009

---From genetics to functional genomics: improvement in drought signaling and tolerance in wheat

Frontiers in plant Science, 2015

----Drought stress: molecular genetics and genomics approaches. Advances in botanical research 57, 445-493

----Hotspots in the genomic architecture of field drought responses in wheat as breeding targets

Functional & integrative genomics 19 (2), 295-309

28

2019

-----Wheat genomics and breeding: Bridging the gap. CABI

  1. Analysis in the method section should be elaborated a bit more.

I would be happy to review it again after revision. 

Author Response

(The authors gave the same response as above.)

Reviewer 3 Report

This manuscript report a work on applying trehalose in the culture solution for evaluating its effects on the physiologic traits and transcriptomic changes in the root of wheat at seedling stage. The design of the experiments was simple and far away from the actual condition, though the analysis was conducted following the normal ways, do not provide much useful informations for revealing the roles of trehalose in improving heat tolerance in the roots of wheat. 

Author Response

(The authors gave the same response as above.)
